# Comprehensive Assessment of Candidate Reference Genes for Gene Expression Studies Using RT-qPCR in *Tamarixia radiata*, a Predominant Parasitoid of *Diaphorina citri*

**DOI:** 10.3390/genes11101178

**Published:** 2020-10-10

**Authors:** Chang-Fei Guo, Hui-Peng Pan, Li-He Zhang, Da Ou, Zi-Tong Lu, Muhammad Musa Khan, Bao-Li Qiu

**Affiliations:** 1Key Laboratory of Bio-Pesticide Innovation and Application of Guangdong Province, South China Agricultural University, Guangzhou 510642, China; changfeiguo@163.com (C.-F.G.); panhuipeng@scau.edu.cn (H.-P.P.); shengfang447@126.com (L.-H.Z.); ouda_IPM@stu.scau.edu.cn (D.O.); tat_tong@163.com (Z.-T.L.); drmusakhan@outlook.com (M.M.K.); 2Engineering Research Center of Biocontrol, Ministry of Education, Guangzhou 510642, China; 3Maoming Branch, Guangdong Laboratory of Lingnan Modern Agriculture, Maoming 525000, China

**Keywords:** *Tamarixia radiata*, RT-qPCR analysis, reference gene, *RefFinder*, gene expression

## Abstract

*Tamarixia radiata* (Waterston) is a predominant parasitoid of the Asian citrus psyllid (ACP), a destructive citrus pest and vector of huanglongbing (HLB) disease in the fields of southern China. To explore the functioning of target genes in *T. radiata*, the screening of specific reference genes is critical for carrying out the reverse transcriptase-quantitative polymerase chain reaction (RT-qPCR) under different experimental conditions. However, no reference gene(s) for *T. radiata* has yet been reported. Here, we selected seven housekeeping genes of *T. radiata* and evaluated their stability under the six conditions (developmental stage, sex, tissue, population, temperature, diet) by using *RefFinder* software, which contains four different programs (*geNorm*, Δ*Ct*, *BestKeeper*, and *NormFinder*). Pairwise variation was analyzed by *geNorm* software to determine the optimal number of reference genes during the RT-qPCR analysis. The results reveal better reference genes for differing research foci: *18S* and *EF1A* for the developmental stage; *PRS18* and *EF1A* for sex, *PRS18* and *RPL13* for different tissues (head, thorax, abdomen); *EF1A* and *ArgK* between two populations; *β-tubulin* and *EF1A* for different temperatures (5, 15, 25, 35 °C); and *ArgK* and *PRS18* for different feeding diets. Furthermore, when the two optimal and two most inappropriate reference genes were chosen in different temperatures and tissue treatments, respectively, the corresponding expression patterns of *HSP70* (as the reporter gene) differed substantially. Our study provides, for the first time, a more comprehensive list of optimal reference genes from *T. radiata* for use in RT-qPCR analysis, which should prove beneficial for subsequent functional investigations of target gene(s) in this natural enemy of ACP.

## 1. Introduction

Huanglongbing (HLB), the so-called citrus greening disease, is a devastating disease impacting citrus trees, caused by “*Candidatus* Liberibacter asiaticus” (*C*Las) [1]. HLB has spread rapidly and severely affected the healthy development of sustainable citrus production. To make matters worse, HLB almost destroyed Florida’s citrus industry, and most citrus-producing regions of the world also suffered a massive blow [1,2,3,4]. Unfortunately, there is currently no cure for HLB [1,3]. The ACP, *Diaphorina citri* Kuwayama (Hemiptera: Psyllidae), is a notorious pest of citrus plants belonging to the Rutaceae family [5], whose entire developmental stages (except for the egg stage) require it feeding on the sap of citrus trees, resulting in stunted shoots and withered branches, and even dieback [6,7]. However, these direct adverse effects alone are insufficient to threaten the healthy development of the citrus industry. Rather, the more significant threat of ACP to global citrus production is that it serves as the principal vector of HLB [5,6,7]. At present, the key to preventing HLB is to cut off its transmission routes; in other words, the effective control of ACP is imperative for stopping HLB’s further spread [5,8].

Insecticides are now the most widely adopted strategy for controlling ACP, but due to their unregulated application and spraying, ACP has developed strong chemical resistance to them [5,8,9,10,11,12]. Hence, devising an alternative control strategy for ACP is very important and essential. *Tamarixia radiata* Waterston (Hymenoptera: Eulophidae) is the most effective natural enemy of ACPand it has been successfully used as a biological control agent of ACP in various countries [13,14,15,16,17,18,19]. Recently, several studies investigated the relationship between venomous proteins of the parasitoid wasp and its host insect prey [20,21]. In addition, serine proteases and serine protease homologs have been identified and characterized by analyzing the information gleaned from genomic and transcriptomic analyses, which provides vital fertile groundwork for decoding the interrelationships between *Pteromalus puparum* and its host [22,23]. Some research has focused on the biological characteristics of *T. radiata* [24], yet few studies have investigated its functional genes [25]. Recently, we sequenced transcriptomic and proteomic data under different temperature conditions and obtained many functional genes (unpublished data). To explore the function of target genes of *T. radiata* and make effective use of this parasitic wasp for the natural control of pests, it is critical to set up an appropriate reverse transcriptase-quantitative polymerase chain reaction (RT-qPCR) protocol that can be used in subsequent functional genomics studies [26]. 

RT-qPCR is not only used to determine gene expression patterns, but also to validate the accuracy of high-throughput transcriptome sequencing (RNA-Seq) data [27,28]. Nonetheless, various factors will influence the accuracy of an RT-qPCR analysis, such as the purity of RNA, the content of complementary DNA, efficiency of the PCR reaction, and its design of primers, to name a few [25,29]. Generally, to obtain the exact expression pattern of a target gene, the determination of reference genes should be standardized. However, much research has shown that the expression level of commonly selected/used reference genes is unstable under different treatment samples [26,30,31,32,33,34,35]. In other words, there seems to be no “universal” reference gene that can be reliably and stably expressed under various experimental conditions. Therefore, experiments should be systematically designed to evaluate the stability of different so-called housekeeping genes of various species under a variety of experimental conditions, for identifying the appropriate reference genes for use in RT-qPCR [36]. Recent studies have documented and tallied the frequency of use of the main reference genes [36]. *18S* ribosomal RNA as a part of the ribosomal RNA, was stably expressed throughout the vast majority of most conditions in various studies, and which was employed its expression level as reference [26]. *Actin* encodes a major structural protein and it is expressed at various levels in many cell types which has been widely used for RT-qPCR analysis. Arginine kinase is the major phosphagen kinase in invertebrates and has been used as a reference gene in previous studies [26]. Elongation factor 1α is involved in protein synthesis and is widely used as a normalizer in insects [36,37]. Ribosomal protein (RP), a principal component of ribosomes, is among the most highly conserved proteins across all life forms and in most of studies, *RPL* and *RPS* family genes was widely used as reference genes [32]. *Tubulin* encodes cytoskeletal structure proteins and its stability was variable under different treatments for the same species in many studies [32].

In our prior work, we had studied several biological characteristics of *T. radiata* [19,38,39]. Yet, the advent of subsequent genome and functional genomics research on *T. radiata* remains limited because no reference genes of this parasitoid have been reported. Accordingly, in this study, we selected the seven most commonly used candidate reference genes—*Actin*(*β-actin*), *RPL13* (ribosomal protein L 13), *β-tubulin*, *18S* (*18S* ribosomal RNA), *ArgK* (arginine kinase), *RPS18* (ribosomal protein S), *EF1A* (elongation factor 1 α)—from the transcriptomes of *T. radiata.* To assess the stability of these genes under six experimental conditions, the *RefFinder* was used which contains four different programs (*geNorm*, Δ*Ct*, *BestKeeper, NormFinder*). Finally, the heat shock protein 70 (*HSP70*), as the target gene, was used to verify our findings. As far as we know, our study is the first to identify stable RT-qPCR reference genes for *T. radiata*, which will enhance our ability to analyze the expression profiles of target genes in future *T. radiata* studies.

## 2. Materials and Methods

### 2.1. Insect Rearing

Parasitoid *Tamarixia radiata* individuals used in this study were initially collected from ACP hosts on *Murraya exotica* plants, in South China Agricultural University, in June 2015. The population was maintained in an incubator housing the ACP nymphs at 26 ± 1 °C and 80 ± 10% relative humidity, under a 14-h-L:10-h-D photoperiod.

### 2.2. Sample Collection of T. radiata 

The reference genes in *T. radiata* were evaluated with respect to its stage of development, sexes, different tissues of the adult female, multiple populations, temperature, and female adults fed on different diets. Each *T. radiata* developmental stage was sampled: lower-instar larvae, higher-instar larvae, pupa, and adult. The numbers of collected samples is shown below: there were 50 lower-instar larvae (consisting of first and second instar), 35 higher-instar larvae (i.e., third and fourth instar), 30 pupae; and 20 adults (10 female and 10 male *T. radiata* adults). Both sexes of *T. radiata* were sampled, namely 20 female and 20 male adults. To collect different tissues, about 40 female adult individuals were dissected for their head, thorax, and abdomen parts, for a total of 120 individuals thus dissected. For the population-level assessment, an indoor population (from a long-term feeding population under indoor conditions) and an outdoor population (collected in the field) were both sampled. For the temperature treatments, a total of 240 (60 individuals for each temperature) individuals female *T. radiata* adults were exposed to 5, 15, 25 or 35 ℃ for 3 h. For the dietary treatments, adults were collected as a single sample maintained on citrus psyllid nymphs or honey, from which samples were collected after 10 days, and 20 individuals constituted a replicate.

Every sample was placed in a centrifuge tube, which contained 50 μL of the TRIzol reagent (Invitrogen, Carlsbad, CA, USA) for the RNA isolation. Each experiment was carried out with three replicates.

### 2.3. Total RNA Extraction and cDNA Synthesis

The RNA of each sample was extracted using TRIzol reagent according to the manufacturer’s directions. The quantity of RNAs were determined with the NanoDrop One spectrophotometer (Thermo Fisher Scientific, Waltham, MA, USA). The total RNA was dissolved in 15–30 µL of ddH_2_O, and the concentrations were as follows: 310.3 ± 11.8 ng/μL (mean ± standard error of the mean (SEM)) for lower-instar larvaes, 403.0 ± 12.1 ng/μL for higher-instar larvaes, 409.0 ± 18.8 ng/μL for pupa, 312.7 ± 14.5 ng/μL for adults, 329.7 ± 22.2 ng/μL for females, 335.0 ± 14.2 ng/μL for males, 222.7 ± 14.1 ng/μL for heads, 258.7 ± 13.3 ng/μL for thorax, 327.0 ± 16.5 ng/μL for abdomens, 324.3 ± 14.0 ng/μL for indoor population, 341.0 ± 15.7 ng/μL for outdoor population, 287.7 ± 16.6 ng/μL for adults under 5 °C, 318.3 ± 7.8 ng/μL for adults under 15 °C, 321.3 ± 11.0 ng/μL for adults under 25 °C, 334.0 ± 17.4 ng/μL for adults under 35 °C, 323.7 ± 14.3 ng/μL for individuals maintained on honey, and 353.7 ± 19.7 ng/μL for individuals maintained on nymphs. The first-strand cDNAs were prepared using 1 μg of RNA from various samples with the PrimeScript RT Kit (Takara, Kyoto, Japan) following the manufacturer’s instructions. Then, the cDNAs were diluted ten-fold before performing the RT-qPCR reactions.

### 2.4. Gene Cloning and Primer Design

Seven reference genes (*β-actin*, *RPL13*, *β-tubulin*, *18S*, *ArgK*, *RPS18*, *EF1A*) frequently used for RT-qPCR investigation in other insect species were assessed. The self-designed primer pairs were based on the Primer Premier 5 Tool, according to recently sequenced transcriptomes for *T. radiata*.

The PCR reaction system contained a 25-μL reaction mixture with the LA Taq DNA polymerase (Takara, Japan). The PCR settings used were those described in a previous study [26]. The purified PCR product was cloned into the pClone007 Blunt vector (TSINGKE, Beijing, China) for sequencing. Finally, the obtained sequences of all reference genes were, respectively, analyzed according to the NCBI database to explore and confirm the function of each gene.

### 2.5. RT-qPCR with SYBR Green

A total 50-μL reaction volume was prepared, consisting of 2.5 μL of each primer, 25 μL of SYBR Green Premix (Takara, Japan), 2.5 μL of diluted cDNA template and 17.5 μL of RNase-free water. The above solution was separated into three technical repeats, each containing about 15 μL of the reaction mixture. All reactions were carried out in the CFX96 real-time PCR system (Bio-Rad, Hercules, CA, USA). The qPCR program included an initial denaturation lasting 3 min at 95 °C, followed by 40 cycles of 95 °C for 10 s and 55 °C for 30 s. A dissociation step cycle (55 °C for 10 s, and then 0.5 °C for 10 s until 95 °C) was added, to conduct a dissociation curve analysis. The RT-qPCR was determined for each gene using slope analysis via linear regression models. The standard curves of each primer pair were drawn based on serial dilutions of cDNA (5^−1^, 5^−2^, 5^−3^, 5^−4^, and 5^−5^). The corresponding RT-qPCR efficiencies (E) were then computed this way: E = (10^[−1/slope]^ −1) × 100.

### 2.6. Determination of Reference Gene Expression Stability

The data for the six experimental conditions were analyzed separately. The stability of each of the seven reference genes was evaluated using *RefFinder* (https://www.heartcure.com.au/reffinder/), which integrates four computational programs: *geNorm* [40], *NormFinder* [41], *BestKeeper* [42], and the Δ*Ct* method [43]. Pairwise variation (Vn/Vn+1) was analyzed between the standardization factors NFn and NFn+1, in *geNorm* software. A value of Vn/Vn+1 ≤ 0.15 meant the *n* reference genes could be considered as the optimal number during the RT-qPCR analysis.

### 2.7. Determination of Gene Expression Levels based on Different Reference Genes

*HSP70s* have already been identified in insects, and research has revealed that *HSP70* is one of the critical predictors of insect tolerance to heat stress, which could protect cells from damage [44]. The study of HSP70 may help elucidate the stress responses of parasitoid wasps and in addition provide insights into their potential application in natural enemy strategies [45]. Hence, given the importance of this gene, the stability of the selected reference genes was assessed using *HSP70* of *T. radiata* as the target gene. This will also lay the foundation for our follow-up research. The forward primer was 5′-AACTTTCGACGTGTCCATCC-3′ and the reverse primer was 5′-ATTTGCGTTCGAACTCCTTG-3′. The *HSP70* expression levels in different temperatures and tissues of *T. radiata* were computed based on their normalization to the two optimal genes and two worst genes.

The relative expression of *HSP70* was calculated by the 2 ^−ΔΔCt^ method [46]. One-way analysis of variance was used for detecting significantly differing *HSP70* expression levels among different *T. radiata* stages and tissues, implemented in SPSS software (v18.0) (SPSS Inc., Chicago, IL, USA).

## 3. Results

### 3.1. PCR Amplification and Performance of Candidate Reference Genes in T. radiata 

In this study, all the reference genes were expressed in *T. radiata* and each amplified a single band (Figure 1). The PCR specificity of reference genes was tested by the melting curve analysis, which showed a single peak (Figure 2). The PCR primer efficiency (E), the correlation coefficient (R^2^), and the linear regression equations are shown in Table 1. Moreover, Figure 3 also presents the standard curve for each reference gene evaluated.

### 3.2. Ct Values of Reference Genes

The expression levels of the seven reference genes were assessed under six experimental conditions, by using cycle threshold (Ct) values. This revealed that, overall, the Ct values of all these reference genes from various experimental groups ranged from 7.5 to 23.12 (Figure 4). Additionally, the Ct value of *18S* was the lowest; hence, it was the most expressed gene under all experimental conditions. From Figure 4, it could be seen that 18S seems to be a relatively stable gene. However, to determine whether it could be used as a reference gene required the following comprehensive analysis. By contrast, most of the other six genes had Ct values of approximately 20, and among them, *Actin* and *ArgK* were the least expressed genes. 

### 3.3. Stability of the Reference Genes under Specific Experimental Conditions

The Δ*Ct* method, *BestKeeper*, *geNorm*, and *NormFinder* were implemented to assess the stability of reference genes in all experimental conditions (Table 2).

*geNorm*. *EF1A* and *RPS18* were deemed optimal genes for the development stage, population and temperature comparisons (Table 2). Both *EF1A* and *β-tubulin* were optimal genes when comparing sexes. *ArgK* and *RPS18* had the best stability in the dietary experiment, while *RPL13* and *RPS18* were evidently optimal genes when analyzing different tissues. In addition, pairwise variation was computed via *geNorm*, to ascertain the least number of reference genes for optimal normalization. Our results suggest that every initial V-value (V2/3) was less than 0.15, which indicated that two reference genes were adequate for optimal normalization under the various experimental conditions explored here (Figure 5).

*NormFinder*. For the development stage comparisons, *EF1A* was the optimal gene. For sex comparisons, *RPS18* showed the best stability. *Actin* had the most stable expression among different tissues. For the two different *T. radiata* populations (laboratory vs. field), *ArgK* was deemed the optimal gene. For the dietary condition, *ArgK* and *RPS18* ranked together as the most stable genes. Additionally, *β-tubulin* was the most stable gene found among the samples exposed to different temperatures. 

*BestKeeper*. This set of analyses revealed that the SD values of *ArgK, EF1A, β-tubulin, RPL13* and *RPS18* were all greater than 1 when comparing the different tissues, precluding their selection as reference genes. In the same way, *ArgK* should not be regarded as a reference gene in development stage comparisons, while *EF1A, β-tubulin* and *RPL13* could not be reference genes for studying sex differences. However, *18S* was designated the optimal gene under conditions of tissue, sex, developmental stage, and temperature comparisons. *EF1A* and *ArgK* were, respectively, the most stable genes for the *T. radiata* population and dietary comparisons.

Δ*Ct* method. This method indicated *PRS18* to harbor the best stability for the tissue, sex and diet experimental conditions, while *18S*, *EF1A* and *β-tubulin* were the most stable genes when considering the development stage, population, and temperature conditions, respectively.

### 3.4. The Overall RefFinder Ranking for Reference Gene Expression Stability

According to *RefFinder*, the integrated reference genes for the development stage, ranked from the most to the least stable, were as follows: *18S*, *EF1A*, *RPS18*, *RPL13*, *Actin*, *β-tubulin*, and *ArgK* (Figure 6A). Likewise, the comprehensive ranking for sex was *RPS18, EF1A, β-tubulin, 18S*, *Actin*, *RPL13*, and *ArgK* (Figure 6B), while that for different tissues was *RPS18, RPL13*, *Actin*, *18S*, *EF1A, β-tubulin,* and *ArgK* (Figure 6C). For the two populations sampled, *EF1A* and *ArgK* were the two most stable genes (Figure 6D), yet *β-tubulin* along with *EF1A* featured the best stability under different temperature conditions (Figure 6E). Interestingly, although *ArgK* was the most unstable gene with respect to developmental stage, sex, tissue, and temperature conditions, it was the optimal gene for when examining the dietary condition of *T. radiata* (Figure 6F). 

### 3.5. Validation of the Selected Reference Genes

The relative expression level of *HSP70* was used to validate the reference genes among the different temperatures and tissues. These results indicated that the expression level of *HSP70* was significantly upregulated at high temperature (35 °C) compared with 25°C; however, *HSP70* expression was about 2.49- and 14.16-fold higher at a high temperature (35 °C) than at 25 °C when normalized to the two most stable reference genes (*β-tubulin, EF1A*) and the least stable reference genes (*RPL13, ArgK*), respectively (Figure 7). This disparity in *HSP70* expression is different. Moreover, expression patterns of *HSP70* were inconsistent among the different tissues when normalized to the two most stable and two least stable reference genes (Figure 8), which meant they have opposite expression patterns.

## 4. Discussion

Quantifying the expression of the target gene by RT-qPCR among various conditions is frequently used in molecular studies [27,47]. For this, housekeeping genes are generally regarded as the most stably expressed among all cell types. Yet, in fact, there are no stably expressed “universal” reference genes under all possible test conditions. Moreover, most of the gene expression studies only use a single reference gene to calculate gene expression levels, which may lead to serious problems and inaccuracies in the data and its interpretation. Therefore, to address this issue, it is both crucial and invaluable to verify the selected reference genes under specific experimental treatments. In recent years, increasingly more studies are reporting on the screening of reference genes in Hemipteran, Lepidoptera, Coleoptera, and Diptera insects [36,37,48,49,50,51,52], but, to date, the reference gene(s) of parasitoid species has been largely overlooked [26,36]. 

The biological characteristics of *T. radiata* were elucidated in our previous studies, including the effects of temperature on its development and its host preference behavior to different stages of ACP, among others [19,38]. However, the reference genes of *T. radiata* are yet to be evaluated, making it difficult to rely upon RT-qPCR to study the gene expression patterns in this parasitoid. Our present study reveals that the transcription level of reference genes depends on the experimental conditions used, which agrees with previous conclusions from a suite of studies that none of the currently available reference genes could serve as a “universal” reference gene [30,31,32,33,34,35,49,51,52,53,54]. For example, we find that *Argk* is the least stable gene in the sex comparison of *T. radiata*, while it was found to be the most stable one in the sex comparison of *Lysiphlebia japonica* [26]. 

Previous studies have suggested that multiple reference genes (at least two reference genes) should be selected to avoid biased normalization in the PT-qPCR analysis [29]. The number of reference genes, required for the normalization of RT-qPCR data, may vary for each experimental treatment [30,31,32,33,34,37,54]. For different developmental stages of *Coleomegilla maculata*, it was recommended that no less than five reference genes be chosen for a reliable normalization, while three reference genes are needed for the sexes [32]. However, using too many reference genes will inconvenience the actual execution and operation of the study in question; hence, it would be ideal to select just two robust reference genes for the reliable normalization. According to the pairwise variations determined by *geNorm*, our study indicates that two reference genes were indeed adequate for each experimental condition we investigated, which is consistent with the study that had suggested two reference genes should be sufficient for target gene normalization [54]. Therefore, based on the combined ranking from *RefFinder*, the most stable genes of *T. radiata* were *18S* and *EF1A* for developmental stages, *PRS18* and *EF1A* for sex condition, *PRS18* and *RPL13* for tissues, *EF1A* and *ArgK* for population-level comparison, *β-tublin* and *EF1A* for temperature treatment, and *ArgK* and *PRS18* for dietary condition. 

In our study, *18S* was the most stable gene across different developmental stages of *T. radiata*, a result consistent with other findings [22,36]. Previously, *18S* has been used as an ideal reference gene under biotic and abiotic conditions in most studies. However, due to its high expression level, the subtle changes in the expression of the target gene may be masked. With this in mind, other genes of ribosomal mechanism could be used, such as RPL and RPS genes [36]. Previous studies have found that RP-encoding genes were optimal reference genes [36]; for example, both *RPL13* and *RPS18* are optimal reference genes among developmental stage, tissue, temperature, and host plant of *Henosepilachna vigintioctopunctata* [54]. Based on our results, however, we infer that *RPS18* is the optimal reference gene when comparing sex and tissue conditions, it being the second-most stable reference gene for the dietary condition of *T. radiata*. According to several studies, *ArgK* is not suitable for use as a reference gene [32,55,56], a generalization supported by our results suggesting that *ArgK* is the least stable gene in developmental stages, sex, tissue, and temperature conditions, yet challenged by our intriguing finding that *ArgK* is the optimal gene under varying dietary conditions, for which *EF1A* is the best reference gene. In addition, *β-tubulin* is widely used as a reference gene [36] when the temperature condition of insects is manipulated, which is consistent with our results. Furthermore, in our study, *Actin* was not stable among any of our six experimental conditions. These findings further reveal that the stability of reference genes is variable under different biological or environmental conditions as well as species. 

In order to further validate the reference genes in *T. radiata*, the relative gene expression levels of *HSP70* was evaluated in different temperatures and tissue. These results further indicate that choosing the best reference gene is crucial for robust RT-qPCR analysis. Otherwise, wrong expression patterns of target genes will affect the sound (true) understanding of target genes due to the illogical misuse of reference genes. 

## 5. Conclusions

In this study, we selected seven housekeeping genes of *T. radiata* to evaluate their stability under six experimental conditions. As far as we know, our report is the first to identify stable RT-qPCR reference genes for the parasitoid *T. radiata*. Hence, these findings should assist in future analyses of the expression profiles of target genes in the *T. radiata*, especially concerning its use in the biological control of ACP to halt the further spread of HLB disease in citrus trees.

## Figures and Tables

**Figure 1 genes-11-01178-f001:**
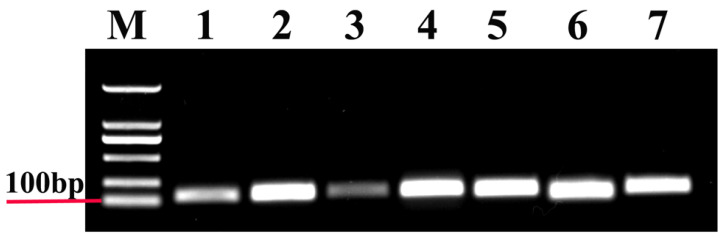
Agarose gel electrophoresis of these seven candidate reference genes. M, Molecular marker. Templates in the polymerase chain reactions (PCRs) were as follows: (1) *18S*; (2) *Actin*; (3) *ArgK*; (4) *EF1A*; (5) *RPL13*; (6) *RPS18*; and (7) *β-tubulin*.

**Figure 2 genes-11-01178-f002:**
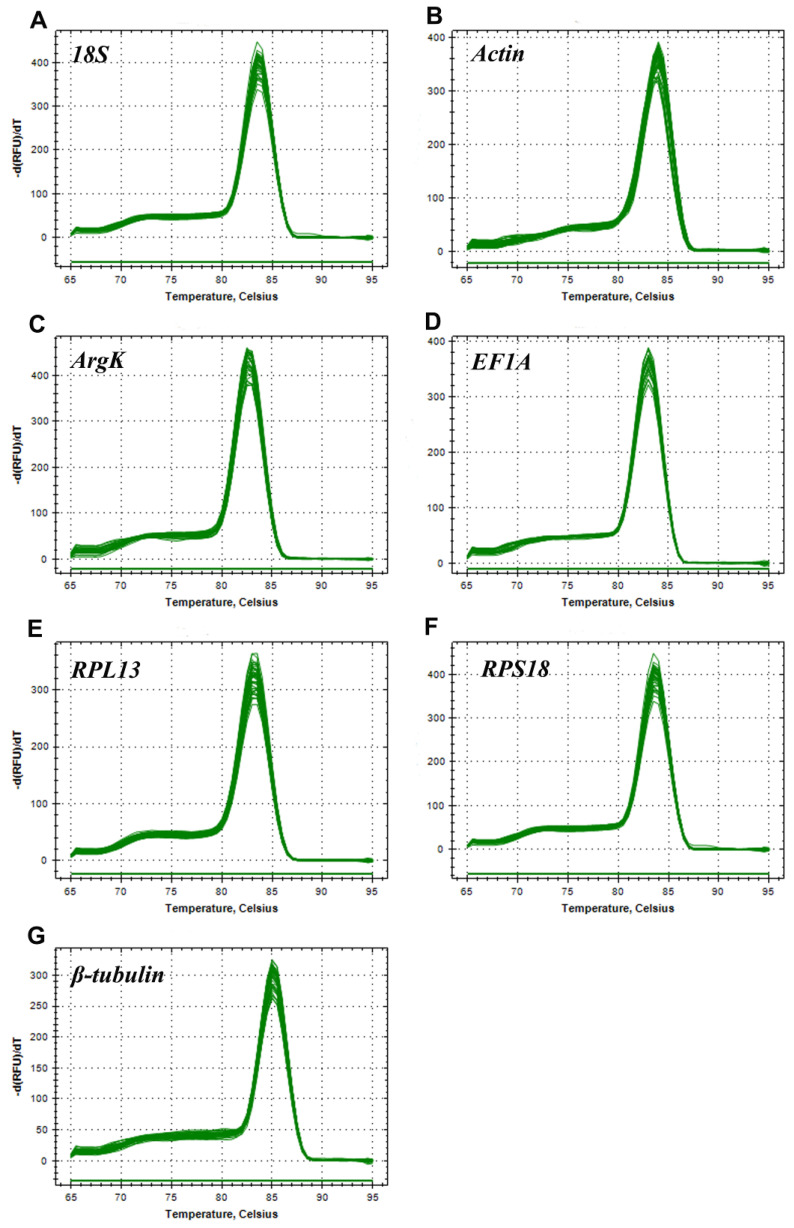
Melting curves of the seven candidate reference genes in *Tamarixia radiata*. (**A**) *18S*, (**B**) *Actin*, (**C**) *ArgK*, (**D**) *EF1A*, (**E**) *RPL13*, (**F**) *RPS18*, and (**G**) *β-tubulin*.

**Figure 3 genes-11-01178-f003:**
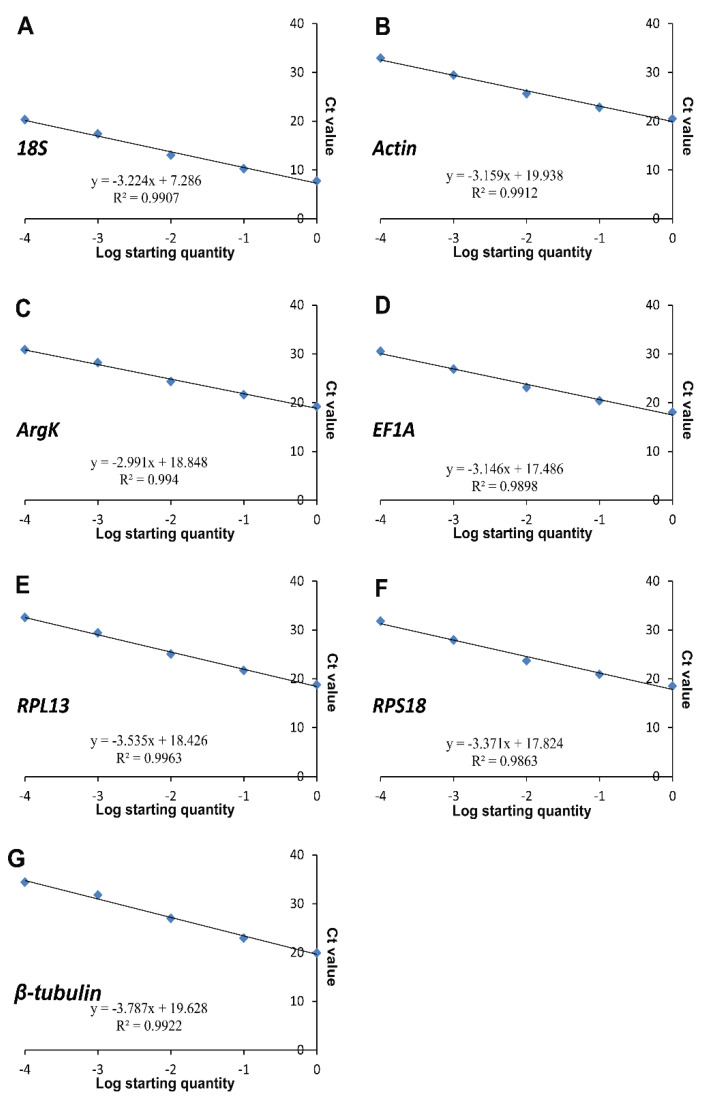
Standard curves of the seven candidate reference genes. (**A**) *18S*, (**B**) *Actin*, (**C**) *ArgK*, (**D**) *EF1A*, (**E**) *RPL13*, (**F**) *RPS18*, and (**G**) *β-tubulin*.

**Figure 4 genes-11-01178-f004:**
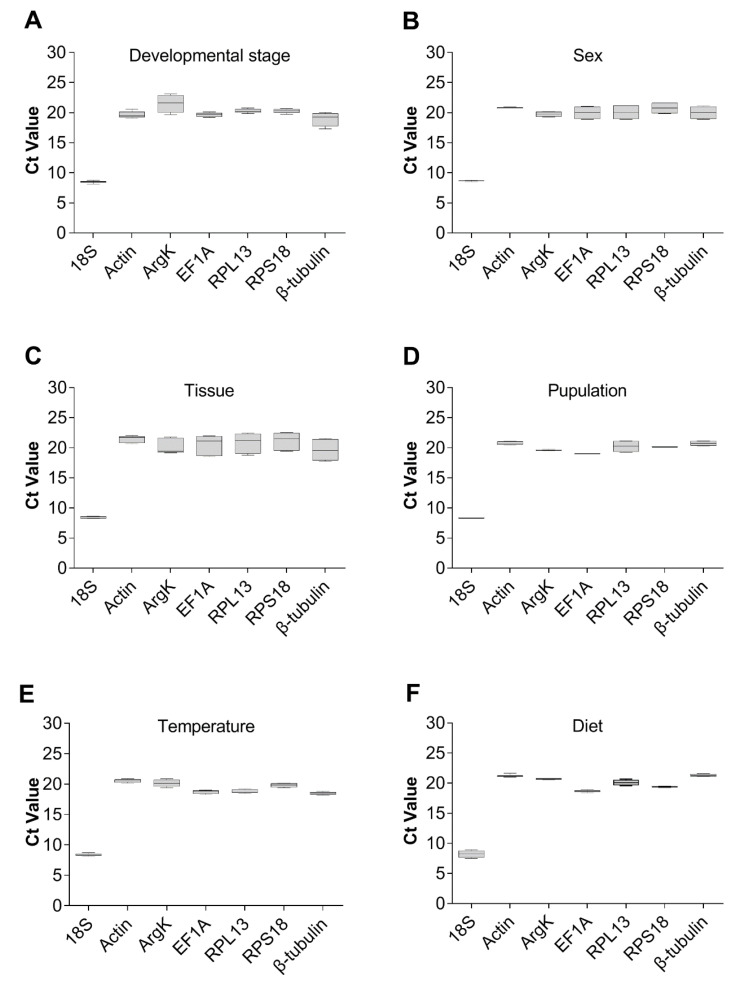
Expression profiles of the seven candidate reference genes in all six experiments for *T. radiata*. The expression levels of the reference genes are shown in terms of the Ct-value for each experimental condition. (**A**) Developmental stage, (**B**) sex, (**C**) tissue, (**D**) population, (**E**) temperature, and (**F**) diet.

**Figure 5 genes-11-01178-f005:**
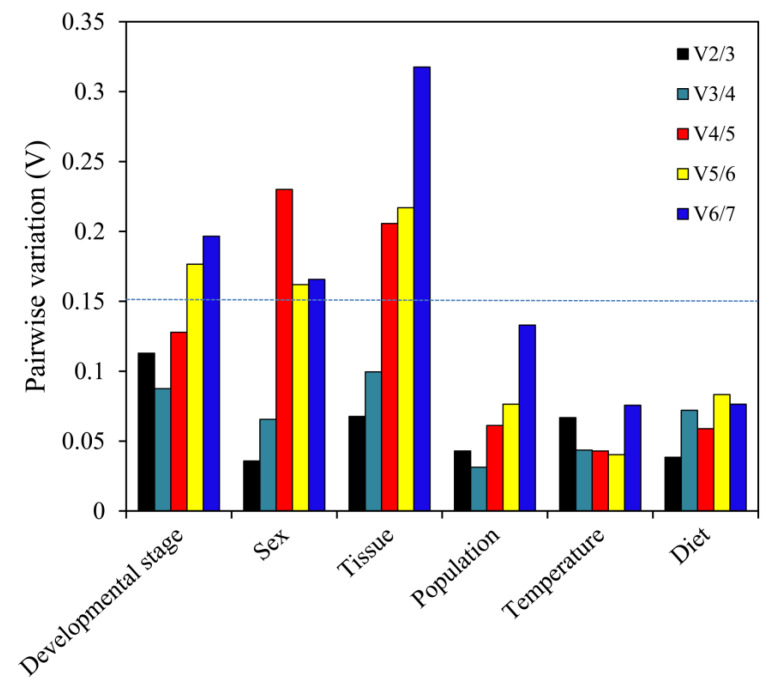
Pairwise variation (V) values using *geNorm* based on different comparisons: developmental stage, sex, tissue, population, temperature and diet.

**Figure 6 genes-11-01178-f006:**
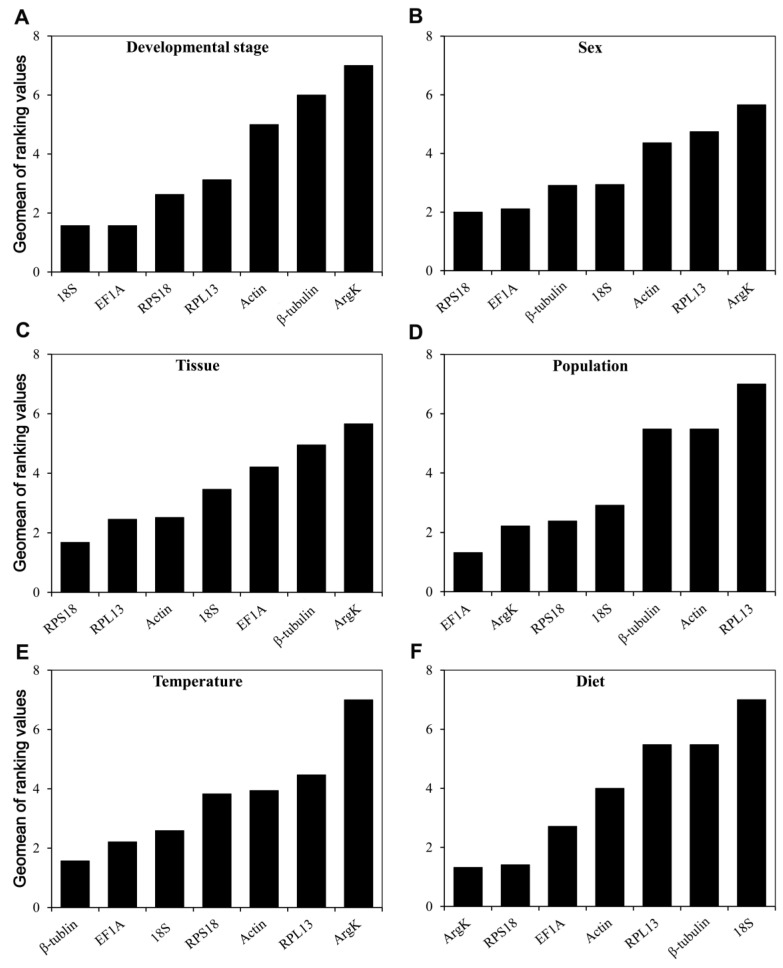
Stability of the seven candidate reference gene expressions in *T. radiata* under different treatment conditions analyzed using *RefFinder*. A lower Geomean value indicates a more stable expression based on *RefFinder*. (**A**) Developmental stage, (**B**) sex, (**C**) tissue, (**D**) population, (**E**) temperature, and (**F**) diet.

**Figure 7 genes-11-01178-f007:**
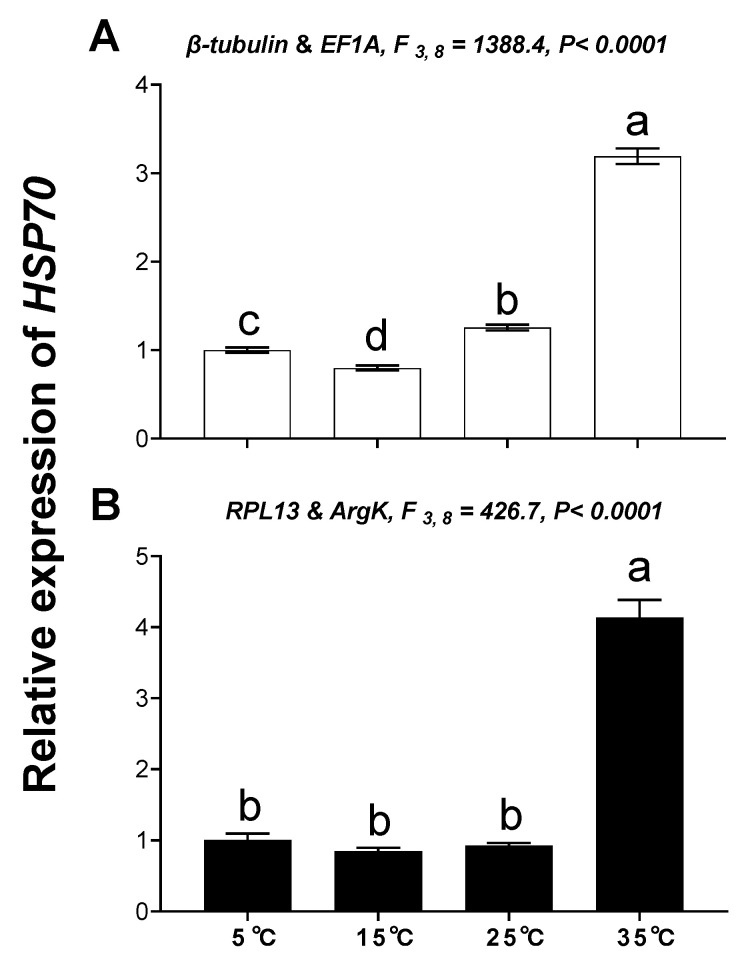
Relative gene expression of *HSP70* in different temperatures. The relative abundance of *HSP70* in different temperatures were normalized to the most stable (**A**: *β-tubulin* and *EF1A*) and least stable (**B**: *RPL13* and *ArgK*) reference genes, respectively. The values are means ± SE (Standard error). The significant differences are indicated by different letters, e.g., a, b, c (*p* < 0.05).

**Figure 8 genes-11-01178-f008:**
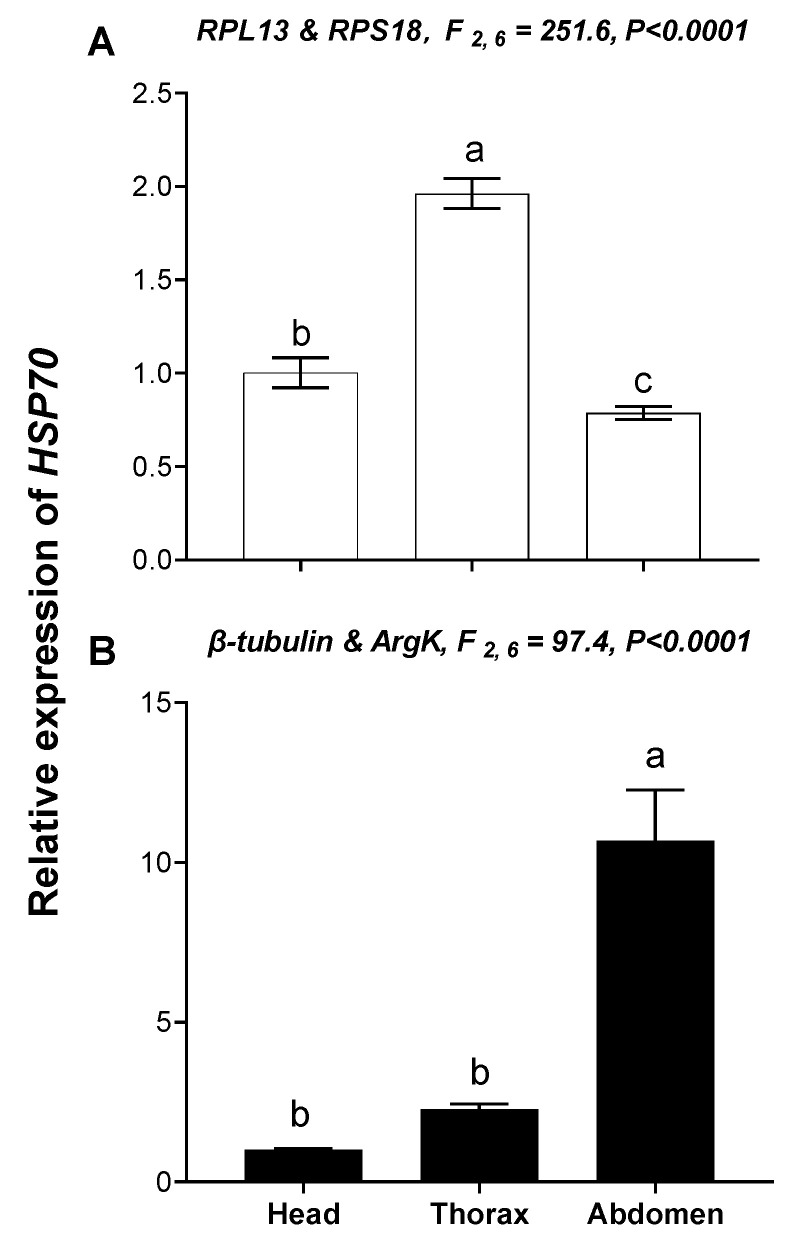
Relative gene expression of *HSP70* in different tissues of *T. radiata*. The relative abundance of *HSP70* in the head, thorax and abdomen were normalized to the most stable (**A**: *RPL13* and *RPS18*) and least stable (**B**: *β*-*tubulin* and *ArgK*) reference genes, respectively. The values are means + SE. Normalized to head. The significant differences are indicated by different letters, e.g., a, b, c (*p* < 0.05).

**Table 1 genes-11-01178-t001:** Reference genes used in this study.

Gene	Primer Sequences (5′–3′)	Length (bp)	Efficiency (%)	R2	Linear Regression
*18S*-F	CTTTCAAATGTCTGCCTTAT	134	104.26	0.9907	y = −3.224x + 7.286
*18S*-R	GCTGCCTTCCTTGGATGTGGT				
*Actin*-F	GTGTGGTGCCATATCTTCTCC	137	107.28	0.9912	y = −3.159x + 19.938
*Actin*-R	ATGGTGGGTATGGGACAGAA				
*ArgK*-F	AGGTCTTGTCCTCGTTGTGG	148	115.94	0.9940	y = −2.991x + 18.848
*ArgK*-R	CCATTAACCGGAATGAGCAA				
*EF1A -F*	TTCCTTCAAGTATGCCTGGG	139	107.90	0.9898	y = −3.146x + 17.486
*EF1A -R*	GAAATCTCTGTGTCCAGGGG				
*RPL13-F*	TAGCCTTTTTGACCGGCAC	90	91.82	0.9963	y = −3.535x + 18.426
*RPL13-R*	GCACAAGGGTCTCAGAAAGG				
*RPS18-F*	ACAAAAGGACATTGTTGACGG	137	97.99	0.9863	y = −3.371x + 17.824
*RPS18-R*	GAACACGGAGACCCCAGTAG				
*β-tubulin-F*	GAGCAGATGCTCAACATCCA	134	83.68	0.9922	y = −3.787x + 19.628
*β-tubulin-R*	GTGCTGTTTCCGATGAAGGT				

**Table 2 genes-11-01178-t002:** Stability of candidate reference genes expression under different experimental conditions calculated by *geNorm*, *Normfinder*, *BestKeeper*, and the Δ*Ct* method, respectively.

Conditions	Rank	*geNorm*		*NormFinder*		*BestKeeper*		Δ*Ct*	
		Gene	Stability	Gene	Stability	Gene	Stability	Gene	Stability
Developmental stage	1	*EF1A*	0.306	*EF1A*	0.054	*18S*	0.12	*18S*	0.66
	2	*RPS18*	0.306	*18S*	0.182	*RPL13*	0.24	*EF1A*	0.68
	3	*18S*	0.35	*RPL13*	0.268	*EF1A*	0.26	*RPS18*	0.73
	4	*RPL13*	0.374	*RPS18*	0.337	*RPS18*	0.27	*RPL13*	0.73
	5	*Actin*	0.492	*Actin*	0.699	*Actin*	0.39	*Actin*	0.91
	6	*β-tubulin*	0.696	*β-tubulin*	1.066	*β-tubulin*	0.88	*β-tubulin*	1.21
	7	*ArgK*	0.907	*ArgK*	1.358	*ArgK*	1.23	*ArgK*	1.44
Sex	1	*EF1A*	0.07	*RPS18*	0.311	*18S*	0.06	*RPS18*	0.69
	2	*β-tubuulin*	0.07	*EF1A*	0.607	*Actin*	0.09	*EF1A*	0.73
	3	*RPL13*	0.098	*18S*	0.626	*ArgK*	0.41	*β-tubulin*	0.74
	4	*RPS18*	0.182	*β-tubulin*	0.629	*RPS18*	0.83	*RPL13*	0.8
	5	*18S*	0.57	*Actin*	0.692	*EF1A*	1.02	*18S*	0.85
	6	*Actin*	0.707	*RPL13*	0.745	*β-tubulin*	1.04	*Actin*	0.88
	7	*ArgK*	0.837	*ArgK*	1.132	*RPL13*	1.12	*ArgK*	1.16
Tissue	1	*RPL13*	0.179	*Actin*	0.218	*18S*	0.14	*RPS18*	0.93
	2	*RPS18*	0.179	*RPS18*	0.514	*Actin*	0.49	*RPL13*	0.99
	3	*EF1A*	0.207	*RPL13*	0.753	*ArgK*	1.08	*EF1A*	1.04
	4	*β-tubulin*	0.308	*18S*	0.779	*RPS18*	1.12	*Actin*	1.1
	5	*Actin*	0.601	*EF1A*	0.821	*β-tubulin*	1.24	*β-tubulin*	1.11
	6	*18S*	0.846	*β-tubulin*	0.863	*RPL13*	1.25	*18S*	1.3
	7	*ArgK*	1.244	*ArgK*	2.205	*EF1A*	1.32	*ArgK*	2.24
Population	1	*EF1A*	0.06	*ArgK*	0.057	*EF1A*	0.01	*EF1A*	0.33
	2	*RPS18*	0.06	*18S*	0.059	*RPS18*	0.06	*ArgK*	0.34
	3	*18S*	0.108	*EF1A*	0.09	*ArgK*	0.08	*18S*	0.34
	4	*ArgK*	0.124	*RPS18*	0.191	*18S*	0.1	*RPS18*	0.35
	5	*Actin*	0.199	*β-tubulin*	0.231	*Actin*	0.27	*β-tubulin*	0.48
	6	*β-tubulin*	0.286	*Actin*	0.502	*β-tubulin*	0.37	*Actin*	0.53
	7	*RPL13*	0.471	*RPL13*	0.926	*RPL13*	0.87	*RPL13*	0.93
Temperature	1	*EF1A*	0.138	*β-tubulin*	0.12	*18S*	0.14	*β-tubulin*	0.26
	2	*RPS18*	0.138	*Actin*	0.123	*β-tubulin*	0.18	*EF1A*	0.28
	3	*β-tubulin*	0.187	*18S*	0.133	*EF1A*	0.22	*18S*	0.28
	4	*RPL13*	0.193	*EF1A*	0.163	*Actin*	0.24	*RPL13*	0.28
	5	*18S*	0.212	*RPL13*	0.188	*RPL13*	0.24	*Actin*	0.29
	6	*Actin*	0.233	*RPS18*	0.254	*RPS18*	0.29	*RPS18*	0.31
	7	*ArgK*	0.322	*ArgK*	0.526	*ArgK*	0.42	*ArgK*	0.54
Diet	1	*ArgK*	0.063	*ArgK*	0.032	*ArgK*	0.08	*RPS18*	0.28
	2	*RPS18*	0.063	*RPS18*	0.032	*RPS18*	0.1	*EF1A*	0.3
	3	*EF1A*	0.1	*EF1A*	0.052	*EF1A*	0.12	*ArgK*	0.3
	4	*Actin*	0.197	*Actin*	0.328	*Actin*	0.16	*Actin*	0.42
	5	*β-tubulin*	0.247	*RPL13*	0.381	*β-tubulin*	0.19	*RPL13*	0.45
	6	*RPL13*	0.335	*β-tubulin*	0.437	*RPL13*	0.46	*β-tubulin*	0.48
	7	*18S*	0.396	*18S*	0.524	*18S*	0.56	*18S*	0.55

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
