# Peer review of "Comprehensive Assessment of Candidate Reference Genes for Gene Expression Studies Using RT-qPCR in Tamarixia radiata, a Predominant Parasitoid of Diaphorina citri"

_genes, 2020, doi:10.3390/genes11101178_

Round 1
Reviewer 1 Report
The paper investigated which commonly used reference gene is reliable for gene expression analysis of the parasitoid wasp Tamarixia radiata using RT-qPCR under different conditions. I think the paper is a solid contribution to the field, while the writing needs to be improved and some analyses need to be specified in more detail.
Comments:
- My major concern is the validation of reference genes using vps. It is not clear why the authors chose that gene, which needs to be clarified. Ideally, the candidate gene for the validation analysis should be a gene of which the expression pattern under different conditions is already known (evidence from references or other previous analyses). Furthermore, if it is possible, the authors need to give the RNA-Seq analysis of the vps gene expression among different developmental stages and tissues to support their qPCR results.
- Line 44, need to give the abbreviation for the Asian citrus psyllid.
- Line 54, D. citri and ACP were randomly used.
- Line 58-61, what are the functions of serine proteases and their homologs in biological control?
- Line 127, need to give the seven genes in the text.
- Figure 2, what are the numbers represent on the X- and Y- axes?
- Figure 4, it should be ct value.
- Figures 7 & 8, what are 'a', 'b', and 'c' represent?
- Line 292, change 'RT-QPCR' to 'RT-qPCR'.
Author Response
We thank for your careful read and thoughtful comments on previous draft. We have finished the revison .
Please see the attachment.

Reviewer 2 Report
Here, the authors have identified multiple reference genes for RT-qPCR analyses in Tamarixia radiata, the parasitoid to the agriculturally harmful psyllid, Diaphorina citri. The authors were able to identify potential reference genes to be used when designing different experiments on T. radiata. Overall, the manuscript brings important information to the field, but is severely lacking in manuscript structure, experimental design, and provided data. Due to these major issues, I suggest this manuscript undergo major revisions before acceptance and will only be providing major comments.
Major comments:
Manuscript structure – The figures are not numbered in the order in which they appear in the manuscript. To help with the flow of the manuscript, the first two paragraphs of the introduction (lines 39-51) need to be combined. Additionally, the fourth and fifth paragraphs need to be combined in the introduction (lines 68-84). Information about the target gene needs to be in the introduction. Figure 2 need axis titles and needs to be divided up by letter, as seen in Figure 4. While Figure 3 has axis titles, it needs to be divided up by letter, as seen in Figure 4. Figure 3 could also be placed in the supplement. The results from each primer set need to be separated in Figure 7/8 and/or the data used to construct Figure 7/8 needs to be provided as a supplement.
Experimental design – The authors chose vacuolar protein sorting-associated protein 13 (Vps) as their target gene, but do not mention the known expression pattern of this gene in Drosophila, only that it is essential from brain homeostasis. Expression patterns in Drosophila can be found using Flybase (Vps: CG2093; FBgn0033194) The authors need to provide either alignments or phylogenetic analyses demonstrating the relationship of Drosophila Vps and Tamarixia Vps. The authors also need to provided additional information as to why Vps was chosen as the target gene. I suggest the authors choose an different target gene that has (1) confirmed differential expression in Drosophila across development and (2) relatively high sequence homology between Drosophila and Tamarixia, suggesting potential orthology.
Data Provided – The authors test multiple potential reference genes for their stability (Ct-wise, and through RefFinder), but do not provide the raw Ct values for every RT-qPCR reaction need to be provided in the Supplement. While having multiple reference genes are important the authors do not compare stability across experimental designs. Figure 4 demonstrates the possibility of the 18S primer set to be used across multiple experiments base on Ct variability. I understand that the RefFinder data support the authors’ claims and suggest the authors strengthen their manuscript by taking a closer look at 18S expression. Additionally, the authors need to provide the concentration of RNA used in their cDNA synthesis reactions, as variation in RNA template concentrations can lead to differing RT-qPCR results.
Author Response
We thank for your careful read and thoughtful comments on previous draft. We have finished the revison .
Please see the attachment.

This manuscript is a resubmission of an earlier submission. The following is a list of the peer review reports and author responses from that submission.